# Oncogenic Mutations of MYD88 and CD79B in Diffuse Large B-Cell Lymphoma and Implications for Clinical Practice

**DOI:** 10.3390/cancers12102913

**Published:** 2020-10-10

**Authors:** Carlo Visco, Ilaria Tanasi, Francesca Maria Quaglia, Isacco Ferrarini, Costanza Fraenza, Mauro Krampera

**Affiliations:** Department of Medicine, Section of Hematology, University of Verona, 37134 Verona, Italy; francescamaria.quaglia@univr.it (F.M.Q.); isacco.ferrarini@univr.it (I.F.); costanza.fraenza@studenti.univr.it (C.F.); mauro.krampera@univr.it (M.K.)

**Keywords:** MYD88 mutations, CD79B mutations, diffuse large B-cell lymphoma (DLBCL), cell of origin

## Abstract

**Simple Summary:**

A diagnosis of diffuse large B-cell lymphoma in our therapeutic era should be implemented by the definition of the cell of origin, additional immunohistochemistry (i.e., BCL2 and MYC), and by fluorescent in-situ hybridization. The next step, suggested by the seminary works we will discuss in this review, will be to implement the definition of sub-categories by the recognition of single gene mutations and pathways that may be targetable by newer drugs. We here describe the impact that MYD88 and CD79B activating mutations, two of the most frequent mutations in several DLBCL subtypes, may achieve in the next future in the diagnosis and therapeutics of such a relevant lymphoma subtype.

**Abstract:**

Diffuse large B-cell lymphoma (DLBCL) is the most common non-Hodgkin’s lymphoma in adults. Despite the recognition of transcriptional subtypes with distinct functional characteristics, patient outcomes have not been substantially altered since the advent of chemoimmunotherapy (CIT) twenty years ago. Recently, a few pivotal studies added to the disease heterogeneity by describing several activating mutations, which have been associated with disease presentation, B-cell function and behavior, and final outcome. DLBCL arises from antigen exposed B-cells, with the B-cell receptor (BCR) playing a central role. BCR-activity related mutations, such as CD79B and MYD88, are responsible for chronic activation of the BCR in a substantial subset of patients. These mutations, often coexisting in the same patient, have been found in a substantial subset of patients with immune-privileged (IP) sites DLBCLs, and are drivers of lymphoma development conferring tissue-specific homing properties. Both mutations have been associated with disease behavior, including tumor response either to CIT or to BCR-targeted therapy. The recognition of CD79B and MYD88 mutations will contribute to the heterogeneity of the disease, both in recognizing the BCR as a potential therapeutic target and in providing genetic tools for personalized treatment.

## 1. Introduction

Diffuse large B-cell lymphoma (DLBCL) is the most common type of non-Hodgkin’s lymphoma. Twenty years ago, gene expression profiling studies allowed the recognition of two main molecular subtypes: germinal center B-cell-like (GCB) and activated B-cell-like (ABC) DLBCLs [1,2]. The most recent World Health Organization (WHO) classification of tumors of hematopoietic and lymphoid tissues has divided DLBCLs into three main subgroups based on morphological and molecular specificities: DLBCL NOS (not otherwise specified), high-grade B-cell lymphomas with or without *MYC* and *BCL2* and/or *BCL6* translocations, and other lymphomas of large B cells [3]. The so-called cell of origin (COO) distinction is necessary for the diagnosis of DLBCL NOS [3], and should be made either by adopting accepted immunohistochemical algorithms [4,5,6] or by gene expression assays [7,8]. Co-expression of MYC and BCL2 was considered a new prognostic marker combination, while mutational landscape was reported to provide further definition with a clinical impact that remained to be determined [3,9,10].

DLBCL may express hundreds of genes of germinal-center origin, with the malignant clone undergoing persistent somatic hypermutation, and often switched IgH (Immunoglobulin Heavy) classes. On the other hand, the activated B-cell-like (ABC) molecular subtype is characterized by chronic active BCR signaling and constitutive activation of the nuclear factor κB (NF-κB pathway) [11]. Most ABC lymphomas have not undergone class-switch recombination, and express IgM, unlike most normal germinal-center B cells and GCB lymphomas [7,8]. Constitutive activation of the NF-κB pathway pushes lymphocytes of ABC lymphomas towards maturation into plasma cells, although further genetic events prevent full differentiation [12]. The activation of the NF-kB signaling pathway in ABC DLBCL is frequently due to oncogenic mutations of *CARD11* or *CD79*, strictly associated with canonical NF-κB and BCR activity, or both [13,14]. Furthermore, somatically acquired *MYD88* mutations, a component of the Toll-like receptor danger pathway, have been implicated in cell survival control through the BCR in this lymphoma subtype [15,16].

More recently, the advent of genome-wide molecular profiling has further defined previously unrecognized subtypes within DLBCL [17,18,19]. The genetic features of newly characterized DLBCL subsets, and their unique mutational signatures, have provided additional information for further DLBCL classification. More importantly, these findings have added new insights into DLBCL pathogenesis, providing a genetic tool to identify patients that may benefit from personalized targeted treatment.

## 2. B-Cell Receptor, What Else?

The B-cell receptor (BCR) is a multiprotein complex bound to the surface of mature B-lymphocytes, which plays a central role in DLBCL pathogenesis and proliferation. The BCR is composed of a surface immunoglobulin (Ig) coupled with a CD79A and CD79B heterodimer, which is required for proper cellular localization, trafficking, and signal transduction [20]. Each B cell is equipped with approximately 105 molecules of identical and unique BCRs, playing an essential role not only for antigen recognition but also for general, antigen-independent, B cell survival [20,21]. Therefore, it is not surprising that several types of B-cell lymphomas, including diffuse large B-cell lymphoma (DLBCL), take advantage of BCR expression and co-opt BCR-triggered signaling pathways to sustain neoplastic proliferation [22]. Both variable and constant regions of surface immunoglobulins, as well as functional CD79A/CD79B complexes, are pivotal to determine the initiation, quality, and amplitude of downstream biochemical responses [23]. The immunoglobulin variable heavy chain (*IgVH*) gene repertoire is skewed in a significant proportion of chronic lymphocytic leukemia and DLBCL cases, as the BCR from different patients displays nearly identical complementary determining region 3 (CDR3) sequences [20,24]. Activated B-cell-like (ABC) DLBCLs can use their BCRs to recognize either self-antigens released from apoptotic cells, or surface N-acetyl-lactosamine glycans, or even specific epitopes present in the Ig heavy chain of the BCR itself [21,22]. While the variable regions of the BCR provide the stimuli to trigger antigen-dependent signaling, the constant ones dictate the signaling cascade’s quality and the transcriptional reprogramming [25]. Naïve B cells co-express IgD and IgM BCRs, which change to IgG, IgE, or IgA constant regions after class switch recombination [22]. Due to somatic mutations in the μ region of the IgH gene, 80% ABC DLBCL cases express an IgM-BCR, which leads to pro-survival and mitogenic signals through the NF-kB pathway [22]. By contrast, most germinal center B-cell-like (GCB) DLBCL patients express an IgG-BCR, whose intracellular signaling favors cell differentiation programs rather than tumor proliferation [22,23]. Regardless of the immunoglobulin class employed, both normal and neoplastic B cells rely on immunoreceptor tyrosine-based activation motifs (ITAM) present in the cytoplasmic tails of CD79A and CD79B to initiate the signaling cascade [25]. Progression of the signaling cascade allows NF-kB members to translocate to the nucleus and initiate the transcription of pro-survival target genes [26]. The signaling emanating from BCR mediates the crosstalk with chemokine receptors, integrins, and co-stimulatory molecules, thus controlling several aspects of the B-cell interactions with the surrounding microenvironment [27,28,29].

In normal B cells, two different signaling modes, termed “tonic” and “active”, have been characterized over the past years. “Active”, antigen-driven, BCR signaling is initiated by the aggregation of multiple BCRs, resulting in full phosphorylation of ITAM tyrosine. Physiologically, antigen-triggered BCR response is transient, and several regulatory mechanisms ensure signaling termination. In contrast, “tonic”, low-level, BCR signals are critical for B cell survival independently of high-affinity antigenic stimulation and require continuous activation of phosphatidylinositol-3-kinase (PI3K) [30]. In the neoplastic setting of ABC DLBCL, the BCR and the NF-kB antiapoptotic pathway are constitutively active as a result of oncogenic mutations of CARD11 and/or CD79, with MYD88 mutations playing an external role in contributing at the same level [15]. As a result, the majority of DLBCLs are BCR-addicted tumors, albeit with substantial differences in terms of signaling quality between the GCB and ABC subtypes [31]. For this reason, BCR signaling has emerged as a therapeutic target in several subsets of B-cell lymphoma, including ABC DLBCL.

## 3. CD79 Mutations in DLBCL

CD79 is a heterodimeric protein comprising two transmembrane subunits, CD79A and CD79B, containing a cytoplasmatic region referred to as ITAM, that represents the signal transduction portion of BCR. After antigen-mediated stimulation of the BCR, ITAM is activated by SRC-family kinases and initiates downstream activation of the NF-kB pathway, involving Bruton’s tyrosine kinase (BTK) and the multiprotein “CBM” (CARD11, BCL10, and MALT1) [12,32]. CD79 is expressed exclusively in mature and immature B cells and the vast majority of B-cell neoplasms, in particular non-Hodgkin lymphomas, with relatively higher expression in mantle cell lymphoma (MCL), DLBCL, Burkitt’s lymphoma (BL), and follicular lymphoma (FL) [33]. The most commonly reported CD79B mutation in DLBCL is a missense mutation of the first ITAM motif (Y196). Hotspot mutations on the BCR CD79B subunit have been identified in about 30% of ABC DLBCLs but only 3% of GCB DLBCLs. These mutations play a central role in BCR activation, enhancing the “chronic active” BCR signaling by preventing BCR internalization, and can inhibit negative regulators, such as LYN, and increase BCR expression (++ IgM) on the B-cell surface [12,25,34]. Gain-of-function mutations of CD79B often co-occur with MYD88 mutations, with significantly higher frequency in ABC DLBCLs, while CD79A mutations are uncommon, ranging from 2.9% to 4% of cases [8,12].

## 4. MYD88 Mutations in DLBCL

The myeloid differentiation primary response 88 (MYD88) is an adaptor protein that interacts with receptors containing a Toll/interleukin-1 receptor (TIR) domain. Once activated, MYD88 induces the activation of the IκB kinase (IKK) complex, with subsequent phosphorylation of the NF-κB inhibitor proteins (IκB), nuclear internalization of the NF-κB subunits, and activation of the canonical NF-κB signaling pathway [35].

Recurrent oncogenic mutations affecting MYD88, described for the first time by Ngo et al. [16], can constitutively activate NF-κB signaling and other associated mitogen-activated protein kinases, providing a survival advantage in B-cell malignancies. Interestingly, MYD88 L265P mutations frequently co-occur in DLBCL patients who also harbor CD79B mutations. The overlap of these alterations is too higher to be coincidental, thus indicating a synergistic genetic effect. However, the mutation itself is not capable of inducing carcinogenesis, and additional genetic events are required for this purpose [36,37,38].

In 2011, Ngo and colleagues demonstrated that MYD88 and associated kinases (IRAK1 and IRAK4) are required for the viability of ABC DLBCLs, finding a notable frequency of MYD88 mutations in non-GCB ABC cell lines [36]. The most common MYD88 mutation consisted of a single nucleotide leucine (CTC) to proline (CCG) exchange at position 265 (L265P) in the Toll/interleukin (IL)-1 (TIR) receptor domain of MYD88, and it was detectable in 29% of non-GCB DLBCL biopsies [16]. MYD88 mutations other than L265P have also been described, but at lower frequencies. Subsequent studies confirmed that MYD88 mutations are peculiar to ABC DLBCLs and rarely found in GCB subtype or primary mediastinal large B-cell lymphomas; nonetheless, they are present in 90% of patients with Waldenstrom’s macroglobulinemia (WM) [39].

The prevalence of MYD88 mutations in DLBCL varies according to the lymphoma localization and is significantly higher (present in approximately 70% of cases) in immune-privileged (IP) site-associated DLBCLs, such as primary central nervous system lymphomas (PCNSLs) and primary testicular lymphomas (PTsL) [40,41], or other large B-cell lymphoma subtypes, as will be further addressed in this review.

## 5. MYD88 and CD79 Mutations and Genetically Defined DLBCL Entities

Recently, whole-exome sequencing (WES) studies have been reported in the attempt to decipher the intricate genomic landscape of DLBCLs. Using whole-exome and transcriptome sequencing of 1001 newly diagnosed DLBCLs, Reddy et al. reported on 35 driver genes necessary for the viability of DLBCL cells [19]. In particular, MYD88, CARD11, EBF1, IRF4, and IKBKB inactivation induced the death of ABC DLBCL cells, confirming that these oncogenes affect cell survival and might represent ideal targets for future tailored therapies [19]. An extended genomic analysis performed by Schmitz and colleagues uncovered four genetic subtypes with different phenotypes, clinical behavior, and prognostic significance, comprising 46.6% of DLBCL cases [18]. Among them, the so-called MCD subtype included 71 out of 574 DLBCL cases and was characterized by high frequency of MYD88 or CD79B mutations (82% of cases), with double concomitant mutations in 42% of cases. Of note, 96% of these cases were ABC DLBCL. Moreover, 56% of the MCD displayed loss-of-function lesions affecting components of the negative feedback loop of the BCR, and presented frequent amplification or gain of SPIB, one of the regulator genes of the ABC phenotype, and high expression levels of BCL2 mRNA [18]. A second study by Chapuy et al. identified with similar methods five groups with different genetic signatures [17]. Among them, the C5 group (64 cases) paralleled the previously described MCD group, being characterized by ABC DLBCLs with MYD88 (50%) and/or CD79B (48%) mutations and frequent 18q/BCL2 amplifications [17]. A third pivotal work by Alkodsi et al. [42] identified 36 somatic hypermutation (SHM) target genes capable of stratifying DLBCLs into four novel genetic subtypes. Among them, the SHM2 subtype was characterized by an abundance of alterations in the BCR signaling pathway and harbored the highest frequency of MYD88 (L265P) and CD79B mutations, in addition to other alterations that characterize the ABC subtype, resembling the results described for MCD and C5 groups, respectively.

Of note, with these studies, it became clear that CD79 and MYD88 mutations were mutually exclusive with other known recurrent alterations occurring in DLBCLs, such as translocations of *MYC* and *BCL2* and/or *BCL6*, as observed in the new WHO category of high-grade B-cell lymphomas. Therefore, MD88 and CD79B mutant DLBCLs represent genetic events that specifically characterize a distinct subgroup of DLBCL-NOS with unique pathological and clinical features [40,43,44]. Such observations have strong clinical implications, both related to the prognostic implications of the two mutations, and more importantly because they provide a biologically driven classification system in DLBCL with potential therapeutic applications.

## 6. Prognostic Significance of CD79 and MYD88 Mutations

The prognostic significance of MYD88 and CD79B mutations in ABC DLBCLs is still a matter of debate due to the retrospective nature of most reports, but a negative effect of the mutations, especially for MYD88 L265P, on patient survival is evident.

A retrospective study published by Fernandez-Rodriguez and colleagues, showed that MYD88 L265P represented an independent poor prognostic factor in a series of 129 DLBCLs patients treated with R-CHOP (rituximab plus cyclophosphamide, doxorubicin, vincristine, and prednisone) or CHOP-like regimens (HR 3.448; 95% CI 1.583–7.509; *p* = 0.002) [45].

Rovira and colleagues assessed the frequency of MYD88 mutations in a cohort of 213 DLBCL patients [46]. MYD88 L265P mutations were identified in 39 patients (18%), whereas 8 (4%) had MYD88 mutations other than L265P. In this study, MYD88 L265P mutation was significantly associated with inferior 5-year overall survival (52%; 95% CI: 36–68%; *p* = 0.05). Interestingly, cases harboring mutations other than L265P were typically nodal DLBCLs and showed a better overall survival (OS), even when compared to MYD88 wild-type cases (75%; 95% CI: 45–100% vs. 62%; 95% CI: 55–70%) [46].

A recent meta-analysis of 40 published studies evaluated the prognostic significance of MYD88 L265P mutation in association with age, sex, clinical stage, IPI (International Prognostic Index) score, and overall survival [47]. This pooled analysis demonstrated that the MYD88 L265P mutation significantly correlated with the immune-privileged (IP) tumor site. Overall, of 2736 DLBCL patients, MYD88 L265P was present in 29% (95% IC: 17.2–44.5%). When PCNSL and primary testicular lymphoma (PTL) were excluded, mutations were found in 16.5% of patients. The mutation occurrence rate significantly correlated with ABC subtype, older age, high-risk IPI score, and impaired survival rate (HR = 3.244; 95% CI: 1.784– 5.826; *p* < 0.001), but not with gender or clinical stage [47].

A competitive risk analysis has shown that mutations affecting MYD88 and CD79B were associated with a higher risk of relapse and progression. Moreover, the mutational status of MYD88 improved the predictive performance of the IPI score, suggesting possible future incorporation of MYD88 mutations to the current prognostic scoring systems [43].

Finally, the MCD subgroup, characterized by the frequent occurrence of both MYD88 and CD79B mutations, was associated with the worst outcome among described genetic subtypes (predicted 5-year OS: 26%) [18], as well as DLBCLs with the C5 genetic signature reported by Chapuy and colleagues or the SHM2 subgroup described by Alkodsi et al. [17,42].

The prognostic significance of isolated CD79B mutations in DLBCLs is still unclear. Although some authors reported that CD79B aberrations may confer a survival disadvantage [19], further studies are needed to evaluate the prognostic role of CD79B mutations in wild-type MYD88 DLBCLs.

## 7. Extranodal Lymphomas of Immune-Privileged (IP) Sites

As mentioned earlier, MYD88 and CD79B mutations often coexist in extranodal lymphomas, particularly in those arising from IP sites, such as primary central nervous system and primary testicular lymphomas. MYD88 mutations are by far the most prevalent in PCNSL (75%), or PTL (71%), while they are quite unusual in nodal (17%) or gastrointestinal lymphomas (11%) [40] (see Table 1).

Chen et al. reported that 43.4% of PTLs harbored CD79B mutations, with a relatively low rate of MYD88 mutations in this series (60%), but with frequent co-occurrence of both mutations [41]. In another series, MYD88 and/or CD79B have been detected in more than 70% of PCNSLs, again with a significantly higher rate of MYD88/CD79B double mutant than observed in other DLBCLs [48]. The MCD group described by Schmitz et al., characterized by the co-occurrence of MYD88 and CD79B mutations, not only displayed a high frequency of extranodal localization (28%) but also shared a variety of genetic alterations with PCNSLs, thus suggesting similar pathogenesis [18].

Therefore, several authors consider MYD88 as a hallmark of IP-DLBCLs [49]. One possible mechanism of MYD88/CD79B-derived lymphomagenesis involves TLR/MYD88 signaling, which might initiate tumor growth in IP sites, characterized by their favorable microenvironments, whereas concomitant CD79B mutations could further promote lymphomagenesis by enhancing the BCR signaling [41].

However, a high frequency of mutations involving this pathway has been reported in extranodal DLBCLs not arising in IP sites, such as intravascular large B-cell lymphomas (IVLBCLs), primary cutaneous lymphoma DLBCLs-leg-type (PC DLBCL-LT), vitreoretinal (PVRL), or primary breast DLBCLs, at variable frequency and mostly with concomitant CD79B mutations [47] (Figure 1).

Indeed, a high prevalence of MYD88 L265P (44%) and CD79B (26%) mutations have recently been reported in IVLBCL [50] and in PC DLBCL-LT (70–80% and 40%, respectively) [51,52]. In primary breast DLBCL, MYD88 L265P and CD79B mutations were detected in 58.7% and 33.3% of cases, respectively [53]. Primary breast and primary female genital tract DLBCLs also express similar biological mechanisms, with MYD88 mutations observed in 71% and 50% of cases, respectively, while CD79B mutations were reported in 37% of primary breast and female genital tract DLBCL cases [54]. Further studies confirmed such remarkable site-specific occurrence of MYD88 mutation [55,56].

## 8. Clinical Implications of CD79 and MYD88 Isolated or Concurrent Mutations

Because the frequency of both MYD88 and CD79B mutations varies significantly among different studies, a uniform consensus regarding their clinical and therapeutic implications has not been reached. Moreover, the lymphoma site of origin (IP site versus systemic) seems to be a main factor when predicting tumor response. From a clinical perspective, further insights on MYD88-driven proliferation would be of great interest to allow therapeutic targeting of this pathway. In a phase I/II trial involving 80 relapsed or refractory (r/r) de novo systemic DLBCL patients receiving ibrutinib monotherapy, five of the nine patients (55%) with gain-of-function mutation in the CD79B BCR subunit (23% of ABC DLBCLs) responded. In particular, four out of five patients (80%) with concomitant CD79B and MYD88 mutations responded to ibrutinib, whereas MYD88-mutated/CD79B wild-type cases were unresponsive (0/7). These observations suggested that MYD88/CD79B double-mutant DLBCLs might represent a distinct entity with intense sensitivity to BTK-inhibitors, probably due to CD79B-dependent BCR activation. The same study also showed that inactivation of TNFAIP3 reduced ibrutinib response [15]. Although MYD88/CD79B double-mutant tumors seem to be hyper-addicted to NF-κb signaling, systemic DLBCLs with mutated MYD88 but without BCR mutations (i.e., CD79B) were not responsive (Figure 2). This observation might be related to the induction by abnormal MYD88 of a “chronic active” BCR signaling through a BCR-independent mechanism, thus mitigating sensitivity to ibrutinib. However, the high number of responses that occurred in ABC tumors lacking CD79B mutations (9/29; 31%), especially in the MYD88 wild-type setting, suggests a possible non-genetic mechanism of oncogenic BCR signaling activation, similar to mantle cell lymphoma and other ibrutinib-sensitive B-cell malignancies [15].

The molecular basis for exceptional responses to ibrutinib in DLBCLs has been explained on cell lines and tumor biopsies by the formation of a multiprotein super-complex formed by MYD88, TLR9, and the BCR, whose presence differentiate ibrutinib responders from non-responders [32]. CARD11 is a signal-responsive scaffold protein that acts during the adaptive immune response. Following antigen-mediated lymphocytes stimulation, CARD11 activates the IκB kinase-beta (IKK-β), which acts as a positive regulator of the canonical NF-κB signaling pathway. CARD11 coiled-coil mutations have been described in approximately 15% of ABC DLBCLs, and several studies have illustrated their contribution in maintaining the “chronic active” BCR signaling, allowing tumors to constitutively activate the NK-κB pathway in a BCR-BTK independent manner [14,62]. As a consequence of this BCR-independency, ibrutinib is consistently ineffective in CARD11-mutated patients, both in systemic and IP-sites originating tumors [15,48,63].

Several investigators have addressed the role of ibrutinib in the treatment of PCNSL patients, considering that ibrutinib is a small molecule that rapidly penetrates the blood–brain barrier (BBB), and, as mentioned earlier, PCNSL is enriched by ABC subtype and mutations of CD79B and MYD88. A phase I study showed that 77% of clinical response rates to ibrutinib in relapsed/refractory PCNSLs were significantly higher than the ORR (overall response rate) of 25% reported for r/r DLBCLs without central nervous system (CNS) involvement. Unexpectedly and contrarily to what has been described in systemic DLBCLs, two of the patients achieving CR (complete response) had mutated MYD88 without CD79B mutations [63], although in another series of PCNSL, responses were much lower (Figure 2) [64]. Furthermore, none of the PCNSL patients with concurrent mutations of MYD88 and CD79B achieved CR, although partial responses were seen. A CR was observed in a patient with double wild-type tumors, confirming what was described by Wilson and colleagues and corroborating the hypothesis of a non-genetic mechanism of oncogenic BCR signaling activation at least in some cases [15]. This study also confirmed that coiled-coil CARD11 mutation conferred resistance to ibrutinib [63]. A French study recruited 52 patients with relapsed/refractory PCNSL or PVRL for ibrutinib monotherapy salvage treatment. The intention-to-treat overall response rate was 52% after two 28-day cycles, with documented activity in the brain, eyes, and cerebrospinal fluid. The mutations in the BCR pathway were determined in 18 patients. Consistently with previous findings, responses were observed in the absence of CD79B and MYD88 (and CARD11) mutations (two complete response (CR), two partial response (PR), and three progression disease (PD)). A mutation in MYD88 with no CD79B mutation was observed in nine patients, and two responded. One patient had an isolated mutation of CD79B and achieved PR. No mutation in CARD11 was observed, and no concurrent mutations in MYD88 and CD79B were observed in this study [64]. Overall, the results of these studies still do not allow a clear correlation between mutational profiles in ABC DLBCL and ibrutinib response, both in PCNSL and systemic tumors, to be established, and further biological and clinical data are warranted (Figure 2). To the best of our knowledge, no studies have yet reported cases of PTL treated with ibrutinib monotherapy. A case report on PC DLBCL-LT has described an impressive response to ibrutinib monotherapy in a chemo-refractory patient that harbored mutations of MYD88 and wild-type CD79B [65].

Unfortunately, patients in the studies above mentioned were enrolled irrespective of their mutational status, making the interpretation of results a little tricky. In vivo and in vitro studies have focused on direct inhibition of various components in the MYD88 signaling pathway, showing intriguing results. As an example, an immunomodulatory molecule that inhibits TLR signaling (IMO-8400) has been demonstrated to affect tumor growth in MYD88 mutant cell line models [66,67].

Several other pathways are implicated with the mutational status of MYD88 and CD79b. Phase I/II clinical trials are investigating the response of different therapeutic targets, in addition to BTK inhibition, in this setting [35]. mTOR inhibition with everolimus produced an overall response rate of 50% in patients with WM [68], while PI3K inhibition with parsaclisib produced overall response rates ranging between 20% and 78% in several mutated lymphoma subtypes [69]. In in vitro assays, enzastaurin, a protein kinase C inhibitor, in combination with BTK inhibition, reduced the proliferation and viability of DLBCL cells both by regulating the PI3K, MAPK, and JAK/STAT pathways and increasing the phosphorylation of the BTK [70]. Patients with DLBCL are currently being recruited into a randomized, placebo-controlled phase III study in which enzastaurin is combined with R-CHOP (NCT03263026) [71].

## 9. Liquid Biopsy and Future Practical Implications 

In clinical practice, the collection of tumor tissue is a highly invasive procedure hampered by the risk of severe complications. The so-called “liquid biopsy” represents a non-invasive method to detect tumor-related aberrations using blood plasma or cerebrospinal fluid (CSF), instead of lymphoma tissue. These fluids contain circulating tumor DNA (ctDNA) that is released by the lymphoma cells when they undergo apoptosis or necrosis. Such DNA becomes recognizable and detectable, especially when it harbors somatic mutations, such as MYD88 (L265P). Recently, highly sensitive techniques, such as the droplet digital PCR (ddPCR), have been employed as analysis strategies for targeted mutation detection in CSF samples [72,73], as an alternative to cerebral or retinal biopsy. A pilot study provided evidence that the MYD88 mutation can be reliably detected by a combination of Sanger sequencing and ddPCR in the cell-free DNA (cfDNA) taken from 1 mL of CSF in patients with CNS lymphomas [72]. Therefore, highly sensitive DNA detection is likely to add value to current diagnostic parameters, especially when the available amount of DNA is limited or difficult to reach by standard procedures.

The high frequency of MYD88 (L265P) in several DLBCL subtypes makes this mutation a perfect candidate for liquid biopsy to enter clinical practice. Other advantages of this technique include minimal residual disease evaluation, detection of clonal evolution over time, and spatial differences between the lymphoma cells. An alternative technique for ctDNA analysis is represented by next-generation sequencing (NGS). The benefit of this technique over ddPCR is the possibility of identifying multiple variants, as shown by two studies in DLBCL patients [74,75].

The future practical implications of ctDNA detection strategies shall be guided by the analysis of MYD88 in liquid biopsies. The role of this method in the diagnostic process (i.e., primary vitreoretinal lymphoma), in monitoring disease progression, and in determining the response to therapy is likely to become available soon in most centers, as we advocate its use in every-day practice.

## 10. Conclusions

Collectively, CD79 and MYD88 mutations often coexist in patients with DLBCL and seem to characterize diseases that share clinical, biological, and pathogenetic characteristics. The recently described DLBCL genetic clusters uniformly recognize the occurrence of these mutations within well-defined, distinct subgroups. As such, this brand new DLBCL category is the most robustly described among different studies, is associated with ABC subtype, contains the majority of cases with extranodal sanctuary sites, and is mutually exclusive with double or triple hit high grade B-cell lymphomas. The presence of abnormalities of these genes, accompanied or not by other well-described recurrent mutations, suggests that such redundant alterations may be implicated in the pathogenesis of the disease and may contribute to the selection of treatment-resistant clones. Furthermore, this DLBCL subgroup is associated with the worst overall prognosis when treated with chemoimmunotherapy. The central role of the BCR pathway in this tumor subtype is witnessed by studies reporting unprecedented responses to targeted drugs in such aggressive lymphomas. While MYD88/CD79B double-mutant ABC might be amenable to targeted therapy by BTK inhibitors such as ibrutinib, further studies are needed to define the best therapeutic strategy for isolated CD79B or MYD88-mutant DLBCLs.

## Figures and Tables

**Figure 1 cancers-12-02913-f001:**
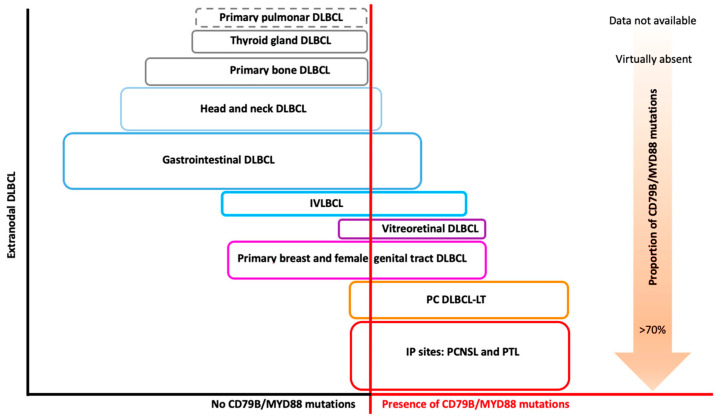
Extranodal DLBCL and CD79B/MYD88 mutations. Conceptual schematic representation of the prevalence of CD79B/MYD88 mutations for each type of extranodal DLBCL (see Table 1 for percentage of mutations prevalence). Each colored box has a different size, approximately corresponding to the frequency of that type of extranodal DLBCL out of the total of extranodal DLBCLs. DLBCL: diffuse large B-cell lymphoma; IP: immune-privileged; PCNSL: primary central nervous system lymphoma; PTL: primary testicular lymphoma; IVLBCL: intravascular large B-cell lymphoma; PC DLBCL-LT: primary cutaneous DLBCL-leg type.

**Figure 2 cancers-12-02913-f002:**
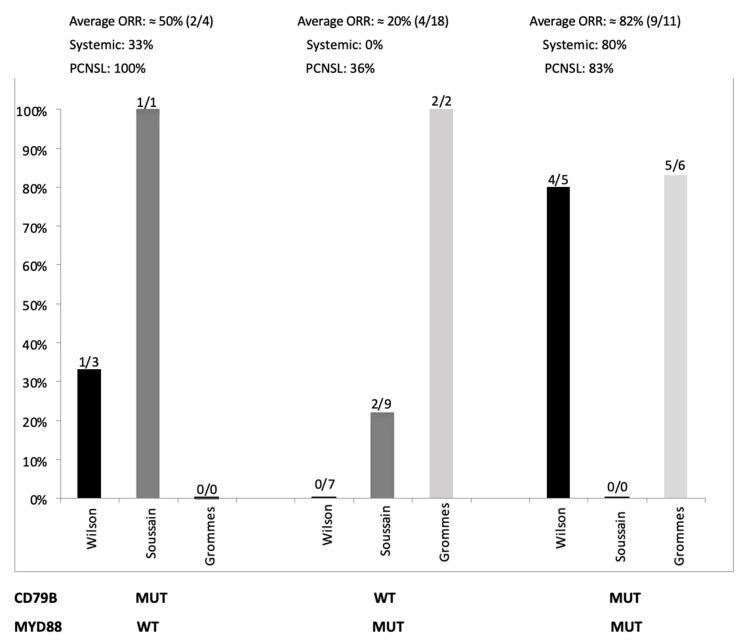
Tumor response rates in three series of patients treated with ibrutinib (Grommes et al. [63], Soussain et al. [64], Wilson et al. [15]) and available mutational status of MYD88 and CD79B. ORR rate for PCNSL refers to the by both Grommes et al. and Soussain et al. [63,64] WT: wild-type; MUT: mutated; ORR: overall response rate; PCNSL: primary central nervous system lymphoma.

**Table 1 cancers-12-02913-t001:** CD79B/MYD88 mutations in DLBCL. DLBCL: diffuse large B-cell lymphoma; GCB: germinal-center B-like; ABC: activated B-cell-like; IP: immune-privileged; PCNSL: primary central nervous system lymphoma; PTL: primary testicular lymphoma; IVLBCL: intravascular large B-cell lymphoma; PC DLBCL-LT: primary cutaneous DLBCL-Leg Type.

CD79B/MYD88 Mutations in DLBCL			
	CD79B	MYD88	Coexisting CD79B and MYD88
DLBCL GCB	0.6–3% [8]	Rare [16]	0.6% [18]
DLBCL Unclassified	6.1% [18]	7.8% [18]	1.7% [18]
DLBCL ABC	21% [57]	29% [16]	11.5% [18]
Nodal DLBCL	3–8% [58]	6–21% [57]	About 9% [57]
**Extranodal DLBCL**			
IP Site: PCNSL	53% [48]	60–79% [40]	About 20% [40]
IP Site: PTL	43.4% [41]	71–77% [40]	About 35% [40]
PC DLBCL-LT	40% [51]	70–78% [51,52]	Unknown [51]
Primary Female Genital Tract DLBCL	37% [54]	50% [54]	About 20% (small sample size) [54]
Primary Breast DLBCL	33.3% [53]	58.7–71% [53,54]	Data not available
Vitreoretinal DLBCL	35% (small sample size) [59]	70% [56]	Data not available
IVLBCL	26% [50]	44% [50]	About 3% [40]
Gastrointestinal DLBCL	<5% [60]	11% [40]	Absent [40]
Primary Bone DLBCL	Absent [60]	Absent [60]	Absent [60]
Thyroid gland DLBCL	Absent [60]	Absent [60]	Absent [60]
Head and Neck DLBCL	Unknown [60]	3.17% (small sample size) [61]	Unknown [60]
Primary Pulmonary DLBCL	Data not available	Data not available	Data not available

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
