# Peer review of "Oncogenic Mutations of MYD88 and CD79B in Diffuse Large B-Cell Lymphoma and Implications for Clinical Practice"

_cancers, 2020, doi:10.3390/cancers12102913_

Round 1
Reviewer 1 Report
The authors comprehensively reviewed and summarized the prevalence, prognostic significance and role of molecular subtyping of diffuse large B cell lymphoma (DLBCL) of MYD88 and CD79B mutations as well as their potential as a predictive biomarker to BTK inhibitors. This review provides useful information on the clinical and potential therapeutic implication of MYD88 and CD79B mutations in DLBCL. I have a few recommendations to improve the readability of article.
Major comment:
- Figure 1 seems to be a little unclear. It needs to be more structured to show the proportion of MYD88 mutations, CD79B mutations and both mutations according to the extranodal sites.
- Figure 2 are difficult to understand. In the Wilson's series, CR and PR status are not described in contrast to other series. ORR rate of PCNSL was from two studies by Soussain and Grommes? Please indicate that clearly. Please include the number of patients after percentage in the ORR sections at the head. In the text describing the data in Figure 2, please mention the first author name, which will allow readers to follow the text and figure more easily.
Minor comment: Please re-check the numbers of patients in Wilson's study. The number of patients with CD79B mutations were 9 in the text but 8 in the Figure 2. The number of patients with CD79B WT and MYD88 Mut 0/7 in the text but 0/9 in Figure 2.
Reviewer 2 Report
It's an interesting, well-written, and high-quality review. Maybe it could be improved in some aspects:
- In the section on clinical implications, a comprehensive review of ibrutinib in this setting is included. The authors could add a perspective on the therapeutic possibilities of other BTK inhibitors and other target drugs related directly or indirectly to MYD88, such as enzastaurin among others (apart from the mention of IMO-8400), and perhaps comment on clinical trials that are being carried out or planning with these drugs.
- Implications in the diagnosis and follow-up. Do the authors consider that the determination of these mutations should be incorporated into the routine diagnosis of patients with DLBCL? Some comment could be added on the possibility of detecting these mutations in liquid biopsy, as a future perspective for diagnosis and / or follow-up.
Minor comments:
- Line 34: BCL2 instead of BCL6.
- Lines 35-36: The wording of the sentence “The so-called cell of origin (COO) distinction is now required for the diagnosis of DLBCL-NOS” must be changed: COO should be made at diagnosis according to WHO 2016, but is not required for diagnosis of DLBCL.
- Figure 2: ORR for WT/WT is 37% (and not 87%).
Round 2
Reviewer 2 Report
I really believe the manuscript has been significantly improved and now warrants publication in Cancers.